# Implementation of a Comprehensive and Personalised Approach for Older People with Psychosocial Frailty in Valencia (Spain): Study Protocol for a Pre–Post Controlled Trial

**DOI:** 10.3390/ijerph21060715

**Published:** 2024-05-31

**Authors:** Mirian Fernández-Salido, Tamara Alhambra-Borrás, Jorge Garcés-Ferrer

**Affiliations:** Research Institute on Social Welfare Policy (POLIBIENESTAR), University of Valencia, 46022 Valencia, Spain; tamara.alhambra@uv.es (T.A.-B.); jordi.garces@uv.es (J.G.-F.)

**Keywords:** integrated care, personalised care, psychosocial frailty, older people, pre–post controlled clinical trial

## Abstract

With ageing, the risk of frailty increases, becoming a common condition that exposes older people to an increased risk of multiple adverse health outcomes. In Valencia (Spain), the ValueCare project develops and applies a value-based care approach that addresses the multidimensional nature of frailty by implementing integrated and personalized care to tackle psychosocial frailty. A pre–post controlled design with a baseline measurement at inclusion, at the end of implementation and a follow-up measurement after 6 months of intervention. In Valencia (Spain), 120 participants over 65 years of age are recruited from primary care centres to receive the ValueCare comprehensive and personalised care plan according to the results and are compared with 120 participants receiving “usual care”. An assessment questionnaire is designed using validated instruments, and a personalised care plan is developed specifically for each participant based on the results obtained. The study protocol has been registered under the ISRCTN registration number ISRCTN25089186. Addressing frailty as a multidimensional and multifactorial risk condition requires the development and implementation of comprehensive assessments and care. In this context, this study will provide new insights into the feasibility and effectiveness of a value-based methodology for integrated care supported by ICT for older people experiencing frailty.

## 1. Introduction

The ageing of the world’s population is indeed a complex and multifaceted phenomenon with both positive and challenging aspects. An ageing population can have far-reaching socioeconomic consequences, for instance, on the care system due to increases in public spending on health and social care [1,2]. As people age, they are at increased risk for chronic diseases, functional impairments, and frailty [3,4,5]. Frailty is indeed considered one of the most problematic expressions of ageing, constituting a risk factor for falls, loss of mobility, cognitive impairment, disability, dementia, hospital admissions, cardiovascular diseases, loneliness, and even mortality [6,7,8]. Research examining the relationship between frailty and loneliness has found strong associations between these two factors, and it suggests a bidirectional relationship [9,10]. Recent studies have confirmed that frail, older people have smaller social networks and higher levels of loneliness [11,12]. Likewise, older people who experience high levels of loneliness have an increased risk of becoming physically frail or prefrail [13]. Similar to frailty, loneliness has also been observed to be closely associated with different negative health outcomes, such as cardiovascular disease, disability, cognitive impairments, depression, disability, and mortality [10,14,15].

In terms of prevalence, in Europe and, in particular, Spain, there is no consensus on the prevalence of frailty. Although studies have confirmed that frailty increases with age and is more frequent in women than in men [16,17,18,19], there are notable variations, depending on the assessment tools, in population variability in terms of geographical location, socioeconomic status, gender, and the presence of chronic diseases [19,20]. A recent study showed that >50% of the population over 50 years of age in Europe is pre-frail/frail, with the overall prevalence of pre-frailty being 42.9% and of frailty 7.7% [20]. In Spain, the most recent studies have shown a prevalence of frailty higher than 25% for the population aged 70 years and older, with significant differences according to geographical areas and being twice as high in women as in men [17,18].

Another issue when studying frailty is that the physical frailty phenotype has received significant attention in the literature, and it is often the most prominent aspect of frailty discussed [21]. However, it is important to note that frailty is a multidimensional concept, and there are other dimensions of frailty that are equally important but may not receive as much attention in the literature. These dimensions include the psychological and social aspects of frailty [22].

In recent years, there has been a growing interest in expanding the understanding of frailty to include these other dimensions and develop more comprehensive and a more holistic approach to managing and preventing frailty and its associated adverse health outcomes. Moreover, the bidirectional relationship between loneliness and frailty, coupled with the multidimensional nature of both constructs, underscores the need for comprehensive strategies to address these issues in older adults [23]. In this sense, the implementation of value-based healthcare principles in the context of frailty lies in addressing frailty from an integrative perspective, reorganising care around patients in an effective and efficient manner supported by digital health solutions [24]. Concretely, among the key principles of value-based care are individualized care plans, shared decision-making, holistic assessments including measurable outcomes, the provision of integrated services, and early interventions. All of these aspects of the value-based care model are essential to addressing the complex interplay of frailty, depression, and social support in older adults [25]. The adoption of value-based care in addressing these conditions among older adults leads to improved health indicators, such as reduced hospitalisations, better mental and physical health, enhanced quality of life, and lower healthcare costs. By focusing on patient-centred, comprehensive, and coordinated care, VBC transforms the healthcare experience for older adults and yields measurable health benefits [26,27].

The literature has shown that social participation is key when developing comprehensive strategies to address frailty, especially frailty understood as a holistic concept including social and psychological aspects [28]. Social participation has been proven to effectively address psychological and social frailty and loneliness [29]. In particular, when developing interventions, it is essential to address the individual dimension, taking into account the individuality of each person, favouring empowerment, and allowing them to manage their own loneliness [14]. In this sense, the literature suggests the need to educate the population about actively investing in their social supports (family and friends) and also emphasises the great importance of boosting the person’s own motivation to actively change their situation [30].

In this regard, the motivational interview technique may be presented as a useful tool to support intervention strategies focused on improving intrinsic motivation and behavioural change among older adults experiencing frailty. Motivational interviewing is a counselling approach developed to help individuals find motivation within themselves to make positive behavioural change [31]. The core principle of motivational interviewing involves engaging in a collaborative conversation with individuals to explore and resolve their ambivalence toward change. Therefore, motivational interviewing may be used for supporting individuals in planning their personal objectives, boosting their motivation, and moving forward with behavioural change that may lead to improved frailty status and loneliness feelings.

This study is framed within the ValueCare project—*Value-based methodology for integrated care supported by ICT*—a research project funded by the European Commission under the Horizon 2020 programme. This project conforms a consortium of 17 partners from 8 European countries. The aim of the ValueCare project is to provide efficient and outcome-based integrated (health and social) care to the population aged ≥65 experiencing frailty, cognitive impairment, and/or multiple chronic conditions, with the objective of improving their quality of life by applying value-based methodologies supported by digital solutions. In this project, study sites in seven European countries are implementing and validating the ValueCare intervention: Valencia in Spain, Rijeka in Croatia, Athens in Greece, Cork/Kerry in Ireland, Coimbra in Portugal, and Rotterdam in the Netherlands. In each of these sites, the ValueCare intervention is aimed to address a specific health condition. In this particular study, the Spanish intervention focused on frailty is presented.

### Objectives

The aim of this study is to evaluate the ValueCare approach implemented in the pilot of Valencia (Spain), whose intervention is based on motivational interviewing, supported by a digital tool, to encourage behavioural change towards greater social engagement and healthier living. This study, using a pre–post controlled design with a study sample of 240 older adults experiencing frailty (120 individuals in the intervention group and 120 in the comparison group), specifically addressed the benefits for older people experiencing psychosocial frailty to be able to evaluate implementation outcomes. We intend to accomplish this objective through the following specific objectives:(a)To compare the benefits of the ValueCare intervention deployed in the Valencia pilot vs. usual care for older people in terms of frailty, loneliness, social support, health-related quality of life, and healthy lifestyle behaviour.(b)To evaluate the benefits of the ValueCare approach deployed in the Valencia pilot centre in terms of reducing the use of outpatient and inpatient health and social care among older people.(c)To evaluate the satisfaction of the target population with the ValueCare intervention deployed in the Valencia pilot centre.

## 2. Materials and Methods

### 2.1. Design

An experimental design involving both intervention and comparison groups will be employed in a controlled pre–post study, aiming to investigate the effects of the intervention by comparing outcomes between the two groups before and after the intervention period.

### 2.2. Study Participants: Inclusion and Exclusion Criteria

The study sample was composed of 240 older adults experiencing frailty (120 individuals in the intervention group and 120 in the comparison group). Participants were randomly assigned to each of the groups. Expecting a 20% loss to follow-up between T0 and T1 (e.g., due to mortality, rehousing, or study withdrawal), we expected to obtain complete data from 96 participants in the intervention group and 96 participants in the control group of each large-scale pilot site (in *n* = 192 study participants with complete data at follow up, equally divided over the intervention group and the control group). We assumed equal standard deviations in the intervention group and the control group, an alpha level of 0.05, and a power of 0.80. For this expected overall sample size and assumptions, with regard to the continuous outcome measures, a difference of 0.23 SD (standard deviation) between the intervention group and the control group can be detected at follow-up.

Participants in the intervention group engaged in a 12-month intervention phase tailored to their needs, receiving a personalised care plan agreed through a shared decision-making process, whereas those in the comparison group maintained their usual care. All participants were evaluated at baseline, after 12 months, and after 18 months.

The inclusion criteria encompassed individuals aged 65 or above, experiencing frailty, residing independently within the community, and affiliated with one of the seven healthcare centres under the Malvarrosa-Clinic Health Department within the Valencia study area. The exclusion criteria involved individuals with cognitive impairments, significant dependency, institutionalisation, inability to provide informed consent, or lack of proficiency in the Spanish language.

Patients who met the inclusion criteria were invited to participate in the ValueCare project, where a comprehensive explanation of their involvement in the project was provided. Patients interested in participating in this study were requested to sign the informed consent form, indicating their voluntary, informed, and explicit consent to participate in this study and permit the processing of their data.

### 2.3. Recruitment and Randomization

The recruitment of the study participants was carried out by social and healthcare professionals from seven participating healthcare centres: Alfahuir Health Centre, Salvador Pau Health Centre, Benimaclet Health Centre, Serrería I Health Centre, Salvador Pau Allende Health Centre, República Argentina Health Centre, and Chile Health Centre. All of these centres belong to the Malvarosa-Clinic Health Department in the city of Valencia.

The general practitioners, who are familiar with their patient’s clinical record, contacted them to ensure that these individuals met the inclusion criteria mentioned above, ensuring their potential eligibility as participants. Once included, participants were requested to complete the baseline assessment. Following the completion of the baseline questionnaire, the allocation of study participants into either the intervention or control group occurred through a randomization procedure. To ensure the concealment of the randomisation sequence, the Oxford Minimization and Randomization (OxMaR) system was employed. This computer-based centralised method ensures proven security measures to prevent bias in the sequence [32]. For safeguarding personal information, every participant was assigned an identification code that corresponds to their group and the specific healthcare centre with which they are affiliated.

### 2.4. Data Collection Process

Data collection and measurement were conducted using an assessment questionnaire that included the International Consortium for Health Outcomes Measurement (ICHOMs) [33] dataset for the older population, as well as additional measurement questionnaires. Each instrument comprising the assessment questionnaire is described in the Section 2.7. Measurement instruments without validated translation into Spanish were translated using the back-translation method to ensure cross-cultural adaptation of the measures.

Researchers inputted the paper-based collected data into the Generic Medical Survey Tracker (GemsTracker) software, chosen for its security measures and capability to collect, submit, and make modifications to the data. Data were collected from participants in three phases: at baseline (T0), 12 months after the end of the intervention (T1), and 18 months (T2).

### 2.5. Description of the Intervention: Design and Implementation

An intervention protocol incorporating aspects like motivational interviewing, social prescription, the transtheoretical model, and person-centred care, was formulated based on an extensive literature review. This review aimed to identify evidence-based interventions addressing psychosocial frailty and establish the methodology to be employed in the intervention process. Moreover, this intervention protocol includes the phases and procedures to be followed to guarantee the effectiveness and sustainability of the intervention. Social and healthcare professionals, including psychologists and social workers, collaborated to design and implement the ValueCare intervention in the Valencia pilot, specifically tailored for individuals aged over 65 experiencing frailty.

All individuals (120) within the intervention group identified as experiencing frailty participated in the intervention phase. After completion of the baseline assessment questionnaire (T0) by the intervention participants, their results were extracted to validate the presence of psychosocial frailty according to the Tilburg Frailty Index [34] and the UCLA 3 Items Loneliness Scale [35].

Participants identified as frail commence the intervention phase by engaging in an initial meeting with social and healthcare professionals. This initial meeting serves a dual purpose: firstly, to present and elucidate the results derived from the baseline questionnaire to the participants, and secondly, to collaboratively design a personalised value-based care plan using a consensus-based co-design approach. This plan was crafted based on the results, preferences, and interests of the participants and will undergo periodic reviews as part of a shared decision-making process.

During the 12 months of the intervention, participants engaged in regular meetings (once a month) with the social and healthcare professionals responsible for their follow-up, during which motivational interviews were conducted. The motivational sessions aimed to achieve several objectives: identifying the psychosocial needs of the individual; assessing the person’s stage of change according to the Transtheoretical Model by Prochaska and DiClemente [36]; fostering motivation to steer the individual towards a readiness for change by assisting in exploring and resolving ambivalences about unhealthy behaviours or habits; and ultimately, collaboratively establishing objectives through a professional–patient negotiation process to guide the transition toward desirable behaviours. The motivational interviews allow a space of trust where the professional’s attitude is one of acceptance and empathy towards the patient’s needs, preferences, and experiences in order to increase and strengthen personal motivation and commitment to change, helping the participant to explore and resolve the ambivalence that arises in people when they have to make decisions that involve behavioural change.

At the end of each monthly motivational session, specific psychosocial objectives focusing on enhanced social participation and the expansion of social networks are agreed with the participant. These objectives will be revisited by social and healthcare professionals in the subsequent meeting for review. The objectives are embedded in the framework of the term “social-prescribing”, where the professional, during the motivational interview, presents to the participant the existing community resources that could potentially enhance their health and well-being [37,38]. These resources are tailored to match the participants’ psychosocial needs, interests, and preferences. Among the community resources introduced may be activities promoted by third-sector entities (associations, foundations, community groups, NGOs…), as well as resources provided by municipally owned institutions such as libraries, art museums, sports centres, theatres, retirement centres, and even the use of a municipality’s green spaces [39,40].

Apart from the motivational sessions with social and healthcare professionals, participants are able to engage in social workshops organized on a monthly basis in each of the health centres participating in the study. These workshops offer opportunities for interaction among participants within the intervention group. Additionally, communication will be facilitated through WhatsApp groups and the dedicated ValueCare APP, allowing further interaction and engagement among participants. In this way, face-to-face relationships are strengthened through information and communication technologies, doubly favouring the active participation of older people in the community, reducing social frailty and improving mental well-being [41].

Specifically, the ValueCare application (Vodafone Innovus, Athens, Greece) [42] is a mobile application with which the participants of the intervention group will be able to interact and through which the achievement of the prescribed goals will be encouraged and healthy lifestyles will be promoted. To ensure that the ValueCare digital solution is adapted to the local context as well as to the needs and interests of older people, informal caregivers, managers, policy-makers, ICT experts, and health and social practitioners, strategies based on collaborative approaches such as co-design are implemented. In this sense, co-design is paramount to providing integrated person-centred care, as it allows for the involvement of all stakeholders in the development of digital health solutions [43]. A total of 2 rounds of co-design were implemented before the ValueCare implementation phase with 212 participants. All co-design rounds included focus group sessions to explore the opinions, perceptions, preferences, and experiences of the target groups around the ValueCare concept and ValueCare solution. A more generic first round aims to define the value-based model and the digital solution to build the concept and the digital solution according to the needs of all stakeholders, presenting and discussing the added value of the value-based concept in today’s society. The second, more specific co-design round aims to define the technical features and properties of the ValueCare solution as well as its involvement and interaction during the implementation phase of the intervention.

The ValueCare application provides each participant with the personalised care plan that has been agreed upon during the professional-participant co-design process. Participants can visualise in the application which are the weekly objectives to be met. Healthcare professionals are in charge of sending these objectives and monitoring the progress of the participants through the Vida 24 web platform (Vidavo S.A., Thessaloniki, Greece. Vida 24 [44] consists of a connected care IT platform that has been operational in Europe for over 10 years and which allows data from multiple sources to be synchronised, personalised, and adapted to specific needs, allowing participants to view the information in the ValueCare Application and professionals to view real-time information on the participants in the Vida 24 platform. In addition, the platform will integrate a virtual coach tool developed by the Fondazione Bruno Kessler [45] that will act as a persuasive chat bot based on dialogue to motivate participants to achieve their objectives and reinforce positive behaviours. In addition, the ValueCare application has different sections where participants can view content in video or text format on existing resources at the community level to maintain an active and healthy lifestyle, tips to increase their social participation, and knowledge on physical frailty, social frailty, and loneliness, among others.

### 2.6. Control Group

During the 12-month intervention phase, the control group, comprising 120 individuals, continued to receive their usual care, visiting primary care centres or hospital care centres to receive attention when needed, as they did before their involvement in the project. Upon completion of the intervention phase, control group participants undergo follow-up evaluation and receive a comparative health outcomes report contrasting T0 and T1. Additionally, a comprehensive guide offering advice on physical, psychological, and social health promotion is delivered to these participants. This guide presents the assets/opportunities available in their local area, encouraging physical activity, healthy eating, and enhancing their active participation in society.

Figure 1 details the flow of participants from recruitment to the last follow-up contact for control and intervention subjects.

### 2.7. Outcomes

First, the screening process for participant eligibility involved the assessment of frailty, dependency levels, and cognitive impairments, according to the inclusion and exclusion criteria.

The FRAIL scale [46] was the tool selected to categorise the level of frailty among participants. It is a commonly used tool that presents a simple and quick assessment that helps identify the presence of frailty based on five key components: fatigue, endurance, ambulation, illness, and weight loss. If the older adult scores ≥1, a high likelihood of frailty is considered to exist [47], and then it meets the criteria for inclusion in the study.

The level of dependency was assessed with the Barthel Index [48]. The Barthel Index is an ordinal scale that measures a person’s ability to perform 10 activities of daily living (ADLs) by providing a quantitative estimate of the subject’s ability to carry out these activities. The ADLs included in the index are eating, personal grooming, toileting, bathing/showering, transferring between the chair and bed, transferring (walking on a smooth surface or in a wheelchair), going up/downstairs, dressing/dressing, stool control, and urine control. Participants were excluded from this study if they scored >60 points on this index.

Finally, cognitive impairments were assessed using the SPMSQ [49], which is a short assessment questionnaire with 10 questions that allows the exploration of different cognitive areas, specifically assessing the functions of orientation, recall memory, concentration, and calculation. If the older adult obtains a score between 0 and 2 on this questionnaire, it is considered a highly suggestive result of cognitive impairment, and the participant is excluded from the study.

The initial assessment (T0) included a comprehensive evaluation of the physical and mental health of the total study sample based on the International Consortium for Outcome Measurement (ICHOM) as a standard set for older people. The main outcome was the health-related quality of life score measured through the Patient-Reported Outcomes Measurement Information System—Global Health (PROMIS-10), which is a 10-item survey that assesses physical health, mental health, satisfaction with social activities and relationships, and quality of life [50]. In particular, to evaluate psychosocial frailty, the Tilburg Frailty Indicator (TFI) was used, which is an instrument that includes the physical, psychological, and social dimensions of frailty [33]. Loneliness was assessed with the UCLA 3-item Loneliness Scale [35]. Health-related quality of life was assessed with the EQ-5D-5L [51], and lifestyle was evaluated in terms of the BMI, smoking, alcohol consumption, physical activity, and nutrition. Moreover, falls were assessed using the previous history of falls and the Visual Analogue Scale for Fear of Falling [52]. Medication adherence was assessed with the Medication Risk Questionnaire (MRQ-10) [53]. Healthcare utilization was assessed with the Modified SMRC Health Care Utilization Questionnaire 18 [54], and finally, socio-demographic data (age, sex, educational level, type of household income, net monthly household income, marital status, and household composition) were also collected.

Complementary to the baseline questionnaire, protein intake was assessed using the Proteiner Screen 55+ [55], and physical performance was evaluated in terms of balance, walking speed, and lower limb strength to get up from a chair using the Short Physical Performance Battery (SPPB) [56].

Table 1 describes the outcome measures used in the evaluation for older people.

### 2.8. Ethics

This research was conducted without any commercial interest on the part of the investigators, the staff of the primary care health centres, or the older people involved in the study. This study received a statement of support based on a previous ethical evaluation by the Human Research Ethics Committee (CEIH) of the Experimental Research Ethics Committee of the University of Valencia (7 May 2020). The content of this study was communicated in a transparent and detailed manner during the recruitment phase, and participation in the study was engaged on a voluntary basis. Participation was consolidated by the individual completion of the informed consent form by each participant. Participants were encouraged throughout the study to contact the investigators if any concerns or questions arose. Informed consent was either collected on paper or electronically. Participants may suspend their participation at any time during this study without disclosing the reasons for their withdrawal. All activities, including data collection and processing throughout the project, comply with ethical principles and relevant national, EU, and international legislation, such as the Chapter of Fundamental Rights of the EU and the European Convention on Human Rights. Provisions of Directive 95/46/EC and the General Data Protection Regulation (proposed in (European Commission, 2012) have been shown to be highly relevant to the protection of research participants and service users. In addition, this study followed the ethical standards and data protection requirements of the GDPR 670/2016.

## 3. Registration and Dissemination

The study protocol has been registered in the International Standard Randomised Controlled Trial Number (ISRCTN25089186; registration date: 16 November 2021).

The investigators aim to disseminate the results of the project in peer-reviewed journals on a regular basis.

## 4. Discussion

This study aims to evaluate the implementation of the ValueCare approach compared to usual care, specifically in the pilot centre in Valencia (Spain), as part of the ValueCare project, focusing on older adults experiencing frailty. In particular, frailty is understood as a multicomponent condition that includes psychological and social aspects in addition to physical ones. The benefits of the intervention will be measured in a wide range of domains for older people: health-related quality of life (HRQoL), activities of daily living, falls, BMI, smoking, alcohol consumption, physical activity, frailty, comorbidities, loneliness nutrition and malnutrition, medication intake, and care utilisation. The outcomes of the implementation will be measured in terms of appropriateness, acceptability, feasibility, fidelity, and costs. A pre–post controlled design will be used to explore the effects of the ValueCare approach on a total of 240 participants belonging to the seven primary care centres that are part of Malvarrosa-Clinic Health as part of the Valencia (Spain) pilot site.

This study is not only highly significant given the high prevalence of pre-frailty and frailty in older people, but also because of the scarcity of comprehensive studies tackling frailty, encompassing its social and psychological dimensions. Indeed, the available literature suggests a scarcity of intervention studies targeting older adults categorised as frail using a comprehensive definition of frailty and who have received personalised treatments [59,60].

Furthermore, the study design within this protocol presents an evaluation of the ValueCare approach in comparison with ‘usual care’ practices in terms of the benefits for older adults experiencing frailty. The benefits of the intervention will be measured in multiple domains apart from frailty: loneliness, health-related quality of life, lifestyle (BMI, smoking, alcohol consumption, physical activity, and nutrition), falls, medication adherence, protein intake, physical performance, and care utilisation.

This study not only offers a comprehensive assessment of frailty but also aims to deliver insights into the effectiveness of a personalised and comprehensive intervention for mitigating and reversing psychosocial frailty by following a pre–post control trial design.

The effects of multidomain interventions related to psychosocial aspects have not been consistent due to the small number of studies examining these outcomes, the scarcity of studies with sufficient statistical power due to inadequate sample size calculation, or even because the beneficial effects on psychosocial health have not been included as primary outcomes [61]. Moreover, this study differs from previous ones as it relies on the value-based care model, with social support being one of the interventions provided within this paradigm. The existing literature on the benefits of social support to prevent or address frailty does not typically use the value-based care approach. The novelty of this approach lies in centring the preferences and needs of each participant, setting goals collaboratively, and supporting them throughout the care continuum.

In addition, to ensure that the design and implementation of the ValueCare approach and ValueCare solution responded to the needs, interests, and preferences of the target groups, the research team implemented collaborative, co-design-based approaches to support adherence to the ValueCare intervention and the use of the ValueCare digital solution by the target groups.

In this sense, this study will address a key gap in the current evidence on the existence of comprehensive interventions. The findings of this study will be disseminated in scientific journals and through scientific and professional conferences.

The proposed study has some limitations, and some challenges are expected to be encountered. Firstly, the recruitment process may be problematic due to the presence of multimorbidities, sensory deficits, transport problems, the influence of other people, fear that the study may harm health, etc. [62,63]. Furthermore, the recruitment process might impose an extra burden on physicians, as they carry the responsibility of enlisting participants from their pool of patients. Thus, research staff can assist in the process by conducting the baseline evaluation after the patient confirms their willingness to participate in the study. Since our target group consists of frail older adults, it is also expected that the participation rate may decrease during the intervention period due to physical or psychological deterioration. Finally, a randomised design is implemented to ensure equal opportunities for subjects to access either the control group or the intervention group. However, this approach might elevate the likelihood of dropout among participants in the control group. This heightened risk can stem from the fact that individuals in the control group solely receive feedback from researchers upon completing evaluation questionnaires (T0–T1).

## 5. Conclusions

Given the rapid growth of the older population worldwide, frailty will place increasing pressure on healthcare systems and will be a major public health care issue in the coming decades. While most studies have addressed frailty by focusing on the phenotypic model of physical frailty, the most current evidence supports the importance of identifying and addressing frailty through a multidimensional approach that takes into account the loss of harmonious relationships between different domains (physical, psychological, and social). New studies incorporating a comprehensive evaluation of frailty must be conducted to introduce innovative interventions that merge social and healthcare aspects. These interventions should aim to yield improved outcomes for older individuals. The literature confirms the fact that psychosocial factors modify the association of frailty with adverse outcomes, with a frail person’s psychosocial resources acting as a protector against adverse outcomes. In this sense, primary care teams need to advance the utilisation of personalised strategies that consider an individual’s social resources, interests, and preferences related to personal activities or social behaviours. These approaches should also incorporate an individual’s ability to self-manage their resources and activities. By fostering the person’s capability to establish and sustain social relationships, as well as encouraging the initiation of social engagement, these strategies aim to enhance social participation. Therapeutic approaches based on motivational interviewing are considered a good starting point in the treatment of psychosocial frailty due to their capacity to strengthen, through a collaborative environment and respect for the autonomy of the person, the intrinsic motivation of the person to enhance self-efficacy towards the initiation and maintenance of behaviour changes towards healthier lifestyles.

## Figures and Tables

**Figure 1 ijerph-21-00715-f001:**
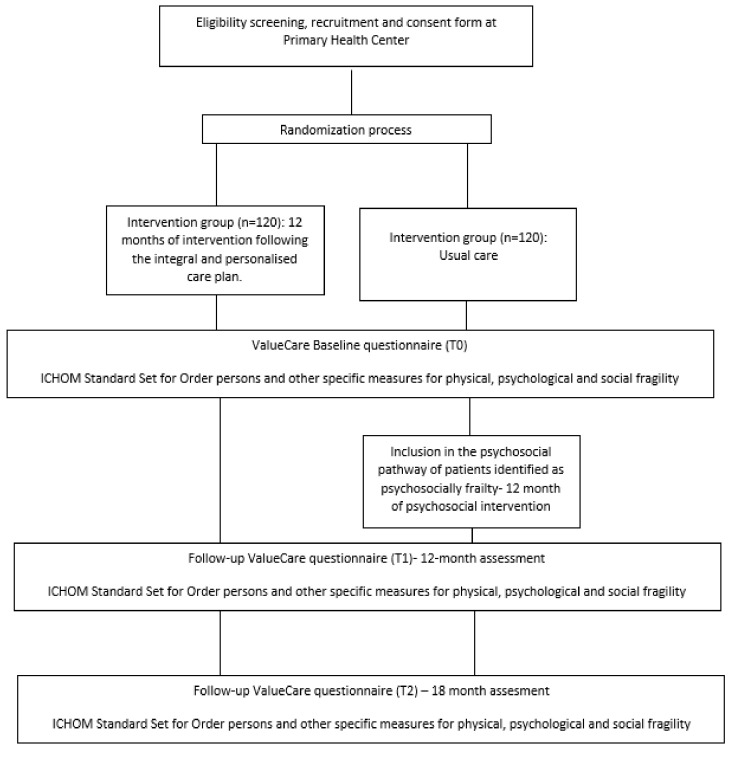
Flow diagram of the participants through the trial.

**Table 1 ijerph-21-00715-t001:** Effectiveness outcomes in older people.

Outcome	Outcome Measure (s)	Methods and Instruments
Health, well-being, and quality of life	Physical HR-QoL	PROMIS-10 [45]
	Mental HR-QoL	
	Self-perceived health	EQ-5D-5L [46]
	Frailty	Tilburg Frailty Indicator [30]
	Comorbidities	ICHOM Older Person Set [29]
	Loneliness	UCLA 3-Item Loneliness Scale [31]
	Activities of daily living	Modified 10-item Barthel Index [43]
	Falls	Visual Analogue Scale for Fear of Falling [47]
Lifestyle behaviour	BMI	ICHOM Older Person Set [29]
	Smoking status	ICHOM Older Person Set [29]
	Alcohol consumption	ICHOM Older Person Set [29]
	Sitting time	One Internal Physical Activity Questionnaire (IPAQ)
	Physical activity	One item of the SHARE-Frailty [57]
	Nutrition and undernutrition	SNAQ + 65 [58]
Medication use	Medication intake	Medication Risk Questionnaire (MRQ-10) [48]
Care use	Care utilisation	Modified SMRC Health Care Utilization Questionnaire [49]

## Data Availability

The data that support the findings of this study are available from the corresponding author upon reasonable request.

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
