# Peer review of "Implementation of a Comprehensive and Personalised Approach for Older People with Psychosocial Frailty in Valencia (Spain): Study Protocol for a Pre–Post Controlled Trial"

_ijerph, 2024, doi:10.3390/ijerph21060715_

Round 1
Reviewer 1 Report
Comments and Suggestions for Authors
A brief summary
Frailty in older adults is a major manifestation of health equity. This study looks at the health risks faced by older people, identifies health outcomes in the older population from a value-based care perspective in a controlled study, and suggests strategies to address the health needs of older people with disabilities. The study's micro-level recommendations can help optimise health management programmes for older people.
Comment 1
In describing the interrelationships between the elements of frailty, depression, and social support in a select group of older adults and the relationship between these elements and health indicators. Can the concept of value-based care as a model be described in detail? And how the results of its action change the health indicators of the older population?
Comment 2
Line 240-256
Are the features of the Value Care app able to support the older population? Is it technically optimised for the elderly population? I would like the authors to elaborate on the friendliness of the app for the elderly population here.
Comment 3
This study deals with the application of digital technology tools to intervene in the frailty of older people. How can we adjust for the problem of data certainty due to insufficient adherence to existing digital technologies, given the relative lack of adherence?
Comments on the Quality of English LanguageNo recommendations
Author Response
Before responding individually to each of the comments, we would like to thank you for taking the time to contribute to the improvement of the quality of this study protocol.
Comment 1: In describing the interrelationships between the elements of frailty, depression, and social support in a select group of older adults and the relationship between these elements and health indicators. Can the concept of value-based care as a model be described in detail? And how the results of its action change the health indicators of the older population?.
Response: Thank you for your comment and we fully understand what you are referring to so we have expanded the information regarding the Value Based Care Model by describing it in detail and justifying how the results of its implementation affect health indicators in the older population. The information included and supported by previous scientific studies is presented in the study protocol on lines 67-80.
Comment 2: Line 240-256. Are the features of the Value Care app able to support the older population? Is it technically optimised for the elderly population? I would like the authors to elaborate on the friendliness of the app for the elderly population here.
Response: Thank you very much for your feedback. The ValueCare digital solution was defined and designed under collaborative co-design processes involving all target groups of the ValueCare intervention. This co-design process was conducted to ensure the creation of a user-friendly digital solution and particularly, a user-friendly interface, with the aim of improving older people's adherence to digital solutions. In the revised version of the paper, in lines 265-280, a detailed explanation of the co-design process for the development of the ValueCare approach and technical solution has been included.
Comment 3: This study deals with the application of digital technology tools to intervene in the frailty of older people. How can we adjust for the problem of data certainty due to insufficient adherence to existing digital technologies, given the relative lack of adherence?
Response: Thank you for pointing this out; it is very interesting. However, the digital solution was designed to complement the value-based care intervention, so the low adherence to the digital tool is not considered a significant problem since the intervention does not depend solely on it. Nonetheless, overall adherence to the intervention was good.

Reviewer 2 Report
Comments and Suggestions for Authors
This protocol is aimed at social implementation with a view to multidimensional support for the elderly and prevention of frailty. It would be meaningful if an unprecedented type of social support could prevent frailty. However, the research question of this study is unclear. There seems to be a lot of research showing that social support prevents frailty, but how is this different? Readers may have little understanding of the significance of this study as there is little description of reviews of previous studies. Lines 377-381 briefly discuss the issues of previous research, but please discuss in more detail how the results of previous research are inconsistent. In the same paragraph there is a point about calculating sample size. However, this protocol also has no description of how to derive the sample size. Please describe how you determined your sample size.
Author Response
Before responding individually to each of the comments, we would like to thank you for taking the time to contribute to the improvement of the quality of this study protocol.
Comment: This protocol is aimed at social implementation with a view to multidimensional support for the elderly and prevention of frailty. It would be meaningful if an unprecedented type of social support could prevent frailty. However, the research question of this study is unclear. There seems to be a lot of research showing that social support prevents frailty, but how is this different? Readers may have little understanding of the significance of this study as there is little description of reviews of previous studies. Lines 377-381 briefly discuss the issues of previous research, but please discuss in more detail how the results of previous research are inconsistent. In the same paragraph there is a point about calculating sample size. However, this protocol also has no description of how to derive the sample size. Please describe how you determined your sample size
Response: We greatly appreciate all the comments and suggestions you have made and have therefore expanded the information available in the study protocol (lines 418-435). Firstly, we have detailed the added value of this study compared to previous studies given the scarcity of previous studies that assess and address frailty in a holistic, comprehensive and personalised way through the implementation of value-based approaches supported by digital solutions, all based on collaborative processes based on co-design with the involvement of all final beneficiaries of the intervention.
In addition, in order to justify the appropriate selection of the sample size for this study, information on sample size calculations has been included (lines 144-153).

Round 2
Reviewer 2 Report
Comments and Suggestions for Authors
The authors have fixed all the matters I pointed out.